# Gastroparesis in Parkinson Disease: Pathophysiology, and Clinical Management

**DOI:** 10.3390/brainsci11070831

**Published:** 2021-06-23

**Authors:** Heithem Soliman, Benoit Coffin, Guillaume Gourcerol

**Affiliations:** 1Centre de Recherche sur l’Inflammation, Université de Paris, Inserm UMRS 1149, 75018 Paris, France; benoit.coffin@aphp.fr; 2Département d’Hépato Gastro Entérologie, Hôpital Louis Mourier, DMU ESPRIT—GHU (AP-HP), 92700 Colombes, France; 3Centre Hospitalo-Universitaire de Rouen, INSERM UMR 1073, CIC-CRB 1404, 76000 Rouen, France; guillaume.gourcerol@chu-rouen.fr

**Keywords:** Parkinson disease, gastroparesis, alpha-synuclein, vagus nerve

## Abstract

Patients with Parkinson disease (PD) experience a range of non-motor symptoms, including gastrointestinal symptoms. These symptoms can be present in the prodromal phase of the disease. Recent advances in pathophysiology reveal that α-synuclein aggregates that form Lewy bodies and neurites, the hallmark of PD, are present in the enteric nervous system and may precede motor symptoms. Gastroparesis is one of the gastrointestinal involvements of PD and is characterized by delayed gastric emptying of solid food in the absence of mechanical obstruction. Gastroparesis has been reported in nearly 45% of PD. The cardinal symptoms include early satiety, postprandial fullness, nausea, and vomiting. The diagnosis requires an appropriate test to confirm delayed gastric emptying, such as gastric scintigraphy, or breath test. Gastroparesis can lead to malnutrition and impairment of quality of life. Moreover, it might interfere with the absorption of antiparkinsonian drugs. The treatment includes dietary modifications, and pharmacologic agents both to accelerate gastric emptying and relieve symptoms. Alternative treatments have been recently developed in the management of gastroparesis, and their use in patients with PD will be reported in this review.

## 1. Introduction

Parkinson disease (PD) is the second most common neurodegenerative disorder, after Alzheimer disease. It affects 2–3% of the population over 65 years and is more common in men [1]. The triad of parkinsonism is defined by motor symptoms that are rigidity, bradykinesia, and tremors [2]. However, the majority of patients with PD reveal a variety of non-motor symptoms, either as a specific complaint or upon specific questioning [3,4]. Gastrointestinal (GI) dysfunction in PD was already described by James Parkinson in 1817 in his first description of the disease [5]. Although historically overlooked [6], interest in GI manifestations has been increasing in the past decades. Several studies revealed five GI features—excess saliva, dysphagia, nausea (mainly related to delayed gastric emptying), decreased bowel movement frequency, and difficulty with defecation—as occurring more frequently in PD patients as compared to aged controls [7,8].

GI manifestations can occur at an early stage of PD and may precede motor symptoms in some cases by several years [9,10]. These disturbances impact the quality of life and are a common reason for emergency room visits and hospitalizations [11,12,13]. Gastroparesis, in particular, contributes to malnutrition and weight loss which is frequent in patients with PD [14]. In addition to its clinical aspect, the idea that PD may have its genesis in the gut has received increasing attention.

In this review, we aim to discuss the pathophysiological changes that might play a role in this GI dysfunction in PD. We will then discuss clinical manifestations linked to gastric dysmotility, namely gastroparesis, the diagnostic criteria of this disorder, and its management, regarding recent data.

## 2. Pathophysiology of GI Dysfunction in PD

Gastric motility and secretory functions are regulated by an extrinsic neuronal network composed of the sympathetic and parasympathetic systems, and an intrinsic neuronal semiautonomous network, the enteric nervous system (ENS). The ENS consists of myenteric (or Auerbach’s) plexus and submucous (or Meissner’s) plexus [15,16]. The myenteric plexus runs between circular and longitudinal muscle layers for the whole length of the gut and primarily provides motor innervation, whereas the submucous plexus plays a role in the control of secretion. The parasympathetic pathway is mainly driven by the vagus nerve, and by the sacral nerves for the distal part of the colon. The extrinsic system cooperates with the intrinsic network, and with the central nervous system. Intramural circuits of the ENS and efferent vagal nerves innervate motor neurons. Excitatory and inhibitory motor neurons drive the motility of the gastric smooth muscle. Interactions between the brainstem and ENS in the form of vago-vagal reflexes determine patterns of normal gastric motor activity [17].

The neuropathological hallmarks of PD are neuronal loss in the substantia nigra, leading to dopamine deficiency, and abnormal α-synuclein accumulation in the brain, with intracellular aggregates leading to the formation of Lewy bodies, or Lewy neurites [18]. The presence of Lewy bodies has been described in the GI tract and especially in the esophagus and colon since 1984 [19]. Mucosal biopsy samples harvested from the colon, stomach, and duodenum, have shown that misfolded α-synuclein is present in the ENS from the early stages in patients with PD and even 8 years before the onset of motor symptoms [20]. Myenteric neurons of the whole GI tract represent one of the earliest sites of α-synuclein accumulation, and this deposition occurs with a rostro-caudal gradient throughout the ENS [21]. High levels of Lewy bodies are also found in the central nervous system and in the dorsal motor nucleus of the vagus nerve (DMV) which has a strong influence on GI motility [22]. The causes of this distribution are unknown, although deposition follows the distribution of visceromotor projection neurons.

These observations led to the “Braak hypothesis”, suggesting that the PD arises within the ENS, presumably triggered by a pathogen from within the gut lumen, and that the disease extends through the vagus nerve to the DMV in the brainstem, and then within the central nervous system [23]. Consistently, several studies revealed an alteration of intestinal permeability which could be the gateway for the disease [24,25]. This hypothesis is sustained by epidemiological studies, from Danish and Swedish registries, reporting that individuals who had undergone full truncal vagotomy were less likely to develop PD than individuals who had undergone selective vagotomy [26,27]. It also prompted investigations to assess enteric α-synuclein deposits, which are far more accessible than the brain as an early biomarker for PD [21]. However, conflicting results have been published, and autopsy studies on 417 patients did not confirm this gradient of deposition and did not find any case in which Lewy bodies and neurites were present in the peripheral autonomic network but not in the brain [28]. Thus, whether the disease spreads from the brain to the gut, or from the gut to the brain through the vagus nerve remains a matter of debate [29,30].

The role of α-synuclein on GI manifestations has not been established. Accumulation of those deposits could lead to a damaged neural network and impairment in gastric motility. Alterations of gastric motility in other synucleinopathies, such as multiple system atrophy, strengthens this hypothesis [31]. However, no neuronal loss has been described, and there has been no association between α-synuclein aggregates and GI symptoms [32,33]. The misfolded α-synuclein could thus play a role in modulating the synaptic pathway. A recent study interested in the spread of misfolded α-synuclein from the DMV to the substantia nigra, and provided the first evidence of an anatomically and functionally defined monosynaptic nigro-vagal pathway that modulates gastric motility [34,35]. This pathway has been shown to be dysfunctional in a rodent model of PD. Dopaminergic inputs to the DMV were then shown to modulate gastric motility, with a gastro-inhibitory response in the healthy model and a different response in PD models [36]. This pathway could thus be the link between the vagus and substantia nigra, and the misfolded α-synuclein in DMV could lead to maladaptive neural plasticity in vagal circuits regulating gastric motility.

Finally, impaired gastric motility in PD is multifactorial, and the ENS should not be considered as the only actor. A recent study revealed that GI dysfunction, specifically constipation, correlates with a reduction in dopamine transporter availability, implying a role for nigral degeneration or change in nigrostriatal dopamine function [37]. Moreover, there is evidence that treatment with levodopa could produce a worsening of gastric emptying, both in healthy volunteers and patients with advanced PD [38]. Alteration of hormone pathways involved in the control of gastric motility has also been documented, especially for cholecystokinin, a hormone known to inhibit gastric emptying [39]. Recent studies focus on the gut microbiome and reported its alteration in patients with PD, but its role in the genesis of the pathology or on GI symptoms remains unclear [40,41].

## 3. Gastroparesis

### 3.1. Prevalence in PD

Gastroparesis is a disorder defined by delayed gastric emptying of solid food in the absence of mechanical obstruction [42]. The main symptoms include early satiety after eating, postprandial fullness, nausea, vomiting, belching, and bloating. Severe forms lead to weight loss and impaired quality of life [43]. The prevalence of gastroparesis in PD has not been formally assessed. Impaired gastric emptying seems to be common reaching 70% to 100% of PD patients in a study using scintigraphy measurement [44]. However, this delayed emptying could be asymptomatic, with subjective symptoms present only in 25% to 45% of patients [45]. Interestingly, a recent study identified a subgroup of PD patients with accelerated gastric emptying [46]. Gastroparesis may occur in early and untreated PD, but its frequency seems to be higher in advanced disease [44,47]. The severity of gastroparesis is also correlated with the severity of motor impairment [48].

### 3.2. Pathophysiology

Delayed gastric emptying is associated with antral hypomotility and in some patients with pyloric sphincter dysfunction. Both mechanisms are caused by neuromuscular dysfunction. Extrinsic excitatory innervation is addressed from the vagus nerve and interacts with the intrinsic nerves of the ENS. In the smooth muscle layer, the interstitial cells of Cajal convey the signal to smooth muscle cells and are regarded as gastric pacemakers. These pacemaker cells do not seem to be altered in PD, suggesting that disturbance occurs either in the vagus nerve or in the myenteric plexus [49]. Alteration in a cholinergic anti-inflammatory pathway has also been demonstrated on an animal model of PD, which could lead to gastric muscular inflammation and muscular macrophage accumulation [50]. This muscular macrophage accumulation in the gastric wall has been described in idiopathic gastroparesis.

The neuropeptide ghrelin is another actor in gastric motility, secreted when the stomach is empty and increases gastric motility [51]. Only one study concerned the role of ghrelin in PD and revealed a decrease in serum concentration as compared with healthy volunteers [52]. Medications used to treat PD, such as anticholinergic or even levodopa, may also delay gastric emptying, which could explain the evolution of gastroparesis in the advanced stages of PD [53,54]. It is also important to remember that delayed gastric emptying might be responsible for medication failure since levodopa needs to reach the small intestine to be absorbed and might contribute to the on–off phenomenon with unpredictable motor symptoms [55].

### 3.3. Diagnostic Criteria

Clinical assessment of the symptoms should be performed with a reproducible and validated scale to allow a better follow-up and to standardize clinical trials on gastroparesis [56,57]. The Gastric Cardinal Symptom Index (GCSI) is to date the best validated score, based on the evaluation of nine items, scored from 0 to 5, (nausea, retching, vomiting, stomach fullness, early satiety, postprandial fullness, loss of appetite, bloating, stomach distension). The global score is then calculated on a range from 0 to 5, and is used to assess the effects of treatment, but not as a diagnostic tool to decide whether a patient should perform diagnostic tests.

Symptoms of gastroparesis are nonspecific and overlap with other sensory or motor upper GI disorders, in particular functional dyspepsia. Patients must first undergo an upper GI endoscopy to rule out any mechanical obstruction. Delayed gastric emptying must then be proven via a specific exam to confirm the diagnosis of gastroparesis. Gastric emptying scintigraphy is the most relevant test for functional and motility investigation. The patient will then take a solid radiomarked meal with a short life radioisotope, ^99m^Tc. The content of the meal is an important factor and has been standardized, with sufficient calories and fat content adapted to Western-style meals; usually this consists of scrambled eggs [58]. The test should last at least 4 h, with image acquisition at 0, 1, 2, and 4 h. Gastroparesis is confirmed if the percent of retention is >60% at 2 h and/or >10% at 4 h [58,59].

The ^13^C gastric emptying breathing test is a validated alternative for scintigraphic measurement and is more accessible with less radiation [60]. The principle of this test is that the rate of ^13^C substrate incorporated in the solid meal is reflected by breath excretion of ^13^CO_2_. The meal incorporates the stable isotope 13C in a substrate such as octanoic acid or spirulina platensis and is ingested after an 8-h fast. Breath samples are then collected before the meal, and at specified times, typically every 30 min, over 4 h [61]. However, this is an indirect test and could be altered by physical activity, by lung or liver disease, or cardiac failure, and by small intestinal absorption [62]. A systematic review on the evaluation of gastric emptying in PD revealed a large discrepancy between scintigraphy study and breath test study, with a higher rate of gastroparesis diagnosed in breath test studies [63]. Finally, the wireless motility capsule has been recently approved by the U.S. Food and Drug Administration (FDA) for gastric emptying measurement and has also been assessed in PD [64]. This single-use orally ingested data recording capsule measures pH, pressure, and temperature throughout the GI tract. It allows measurement of the transit time in the stomach, in the small intestine, and the colon [65]. However, the capsule does not seem to exit the stomach with the meal, as it is a large non-digestible object, but rather with powerful antral contractions of the migrating motor complex which aim to clear the stomach of indigestible material [66].

## 4. Treatment

### 4.1. Dietary Modifications

Therapeutic strategy first relies on dietary modification and is generally used for all patients. Patients are recommended to eat small meals and to avoid foods high in fat and indigestible fibers [67]. A small-particle-size diet has been shown to reduce upper GI symptoms in diabetic gastroparesis [68]. Thus, snacking and more frequent meals to maintain caloric intake are needed. Caloric liquids such as soups are also often well tolerated and recommended. In severe cases, vitamin deficiencies should be detected and supplemented. Rarely, feeding tube or parenteral nutrition can be necessary [69].

### 4.2. Pharmalogical Treatment

Most of the medical treatments used for gastroparesis have not been validated in the specific context of gastric dysmotility due to PD. Prokinetic drugs, and in the first step peripheral dopamine antagonist drugs, are the most commonly used medication. D2 receptor antagonists that cross the brain–blood barrier, including metoclopramide, are contraindicated in PD. By contrast, domperidone is a D2 receptor antagonist acting peripherally as it does not cross the brain–blood barrier, and it may be used to accelerate gastric emptying and relieve nausea and vomiting [70]. Of note, domperidone is associated with cardiac arrhythmia risks and is thus not approved by FDA [71,72]. However, recent data are reassuring on the safety profile of the drug used in the right settings, and domperidone should be considered as an option in gastroparetic PD patients [73]. Motilin receptor agonists, including erythromycin and azithromycin, are not appropriate for extended use owing to drug interactions (especially for erythromycin), to QT prolongation, and to their association with tachyphylaxis with loss of efficacy over a few weeks [74]. A selective 5-HT_4_ receptor agonist, prucalopride, lacking cardiac side effects, is yet approved for the treatment of constipation and has been shown to improve gastric emptying in small open labeled studies in PD [75,76]. One small study reported improvement in gastric emptying with nizatidine, a histamine H2-receptor antagonist, in patients with PD [77]. This drug could also be used to treat some of the reflux symptoms associated with gastroparesis [78]. Finally, ghrelin antagonists such as relamorelin are being assessed as potential prokinetic agents and seem to be effective in improving symptoms and gastric emptying in patients with diabetic gastroparesis in two phase 2 trials [79,80].

Although proton-pump inhibitors (PPI) are often used to treat reflux symptoms that result from gastroparesis, they have been shown to delay gastric emptying and should thus be stopped when possible [81]. Moreover, some studies documented an association between long term use of PPI and cognitive decline even if controversial data have also been published [82,83]. Treatments used in the management of constipation, which is frequent in PD and can be associated with gastroparesis, can also impact gastric emptying. Bulk-forming products, such as psyllium or increasing dietary fiber may delay gastric emptying and cause bloating [84,85]. Osmotic laxatives, primarily polyethylene glycol, should be favored in the context of constipation associated with PD [86].

In case of failure of prokinetics, treatments addressing nausea and vomiting have been used in refractory gastroparesis, but they cannot be recommended in the context of PD. Commonly prescribed agents include prochlorperazine or chlorpromazine, but these treatments are contraindicated as they can worsen the evolution of PD by their action on the central nervous system [87]. Ondansetron, a 5-HT3 receptor antagonist, is considered a reasonable second-line medication in refractory gastroparesis [67]. This treatment is currently assessed as a potential target for psychosis and dyskinesia associated with PD, but its impact on nausea and vomiting in PD has not been specifically evaluated [88,89]. However, the association of ondansetron with apomorphine leads to several adverse effects (sedation, decreased blood pressure) and is contraindicated [90]. The impact of aprepitant, an NK-1 receptor antagonist used to treat chemotherapy-induced nausea, has still not been demonstrated in gastroparesis [91].

### 4.3. Interventional Techniques

Several instrumental techniques are now available for patients who do not respond to medical treatment. In some patients, gastroparesis is associated with pyloric sphincter dysfunction, and endoscopic therapies targeting the pylorus have thus been assessed. Botulinum toxin injections in the pyloric sphincter may alleviate gastroparesis, with data also presented in patients with PD [92]. However, two double-blinded studies failed to show improvement with this technique compared with placebo [93,94]. It may provide temporary relief, but not sustained improvement, lasting on average 3 months. Endoscopic pyloric dilation has been less commonly evaluated in gastroparesis, and not in PD, but could also allow a temporary relief in some patients [95]. Recently, gastric endoscopic pyloromyotomy has been developed for refractory patients, and reveals improvement in gastric emptying and symptomatic scores, with a more sustained relief in 66% of patients at 1 year [96,97]. This technique also has not been assessed in PD, and controlled trials are still missing.

Another approach that should not be forgotten is to circumvent the inconsistency in drug absorption that may result from gastroparesis. Other processes may interfere with response to levodopa and might be improved with these strategies, such as hiatal hernia, Helicobacter pylori infection, or small intestine bacterial overgrowth [98,99]. A variety of non-oral approaches to antiparkinsonian drug administration may be employed. Levodopa could be administered via transdermal patch, subcutaneous and sublingual apomorphine, or liquid infusion [55,100]. Deep brain stimulation of subthalamic nuclei may be required in PD. This technique has been shown to accelerate gastric emptying and to relieve other GI dysmotility symptoms, such as constipation [101,102].

## 5. Conclusions and Future Prospect

Gastroparesis is a frequent disorder in PD patients, and may lead to impaired quality of life, weight loss, and malnutrition. It may also impact drug absorption, and worsen the course of PD.

Much progress has been made in recent years in understanding the pathophysiology of digestive involvement in PD, with a growing role of α-synuclein deposits in the ENS and demonstration of its spreading through the vagus nerve, which interacts with the substantia nigra, and impacts gastric motility. New techniques are being developed to obtain adequate endoscopic biopsy samples from the neuromuscular layers of the stomach and the duodenum. These samples could help evaluate the pathological status of the ENS. Evaluation of pyloric dysfunction with specific endoscopic technique also appears as a promising strategy. Correlation between histological findings, new endoscopic technique evaluations, and treatment outcome could help personalize therapeutic strategy.

Gastroparesis can occur at a very early stage of PD and should be identified promptly and treated. Clinicians should also pay attention to its evolution at each evaluation, with symptomatic and nutritional evaluation. Prokinetics, including domperidone, and dietary modifications are the first line treatments. Newly developed prokinetic drugs require, however, larger validation trials in the context of PD. Endoscopic treatments are currently being developed, and may represent an alternative therapeutic strategy in the future to improve symptoms and gastric emptying. Whether acceleration of gastric emptying leads to a better control of PD symptoms remains, however, to be firmly established.

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
