# Peer review of "Gastroparesis in Parkinson Disease: Pathophysiology, and Clinical Management"

_brainsci, 2021, doi:10.3390/brainsci11070831_

Round 1

Reviewer 1 Report

This is a clearly written review of gastroparesis in Parkinson’s disease. The extensive literature search and logical discussion of findings are beneficial to readers. It is important to draw attention to this component of Parkinson’s disease

Author Response

We are greatful for the reviewer's comment.

The positive opinion of reviewer 1 did not require revision.

Reviewer 2 Report

  1. The abstract should be revised to include or specifically describe the article's highlights/uniqueness.
  2. several references are not in journal format (eg. ref. 16, 46,97)
  3. heading 4 "Treatment" should be subdivided into sub-headings to clearly describe different treatment options.
  4. It is encouraged to incorporate a separate heading "future prospect" or "conclusion and future prospect.”

Author Response

Comment 1: The abstract should be revised to include or specifically describe the article's highlights/uniqueness.

We added two sentences in the abstract to highlight the high prevalence of gastroparesis in PD, and to describe the specificity of this article to discuss the management of gastroparesis in the context of PD.

Comment 2: Several references are not in journal format (eg. ref. 16, 46, 97)

We have corrected the cited references in order to fit the journal format. We also checked the other references that were in the appropriate format.

Comment 3: Heading 4 "Treatment" should be subdivided into sub-headings to clearly describe different treatment options.

We acknowledge that the “Treatment” section was quite developed. We added three sub-headings to clearly describe the different steps of the therapeutic strategy, with first dietary modification; then pharmalogical treatment and lastly the interventional techniques.

Comment 4: It is encouraged to incorporate a separate heading "future prospect" or "conclusion and future prospect.”

We changed the heading of the conclusion, which became “conclusion and future prospect”. We thus added a paragraph on the future prospects in the field of gastroparesis in PD, after the conclusion.

Round 2

Reviewer 2 Report

The authors have revised the manuscript and satisfactorily responded to the comments.